# Secretome Analyses Identify FKBP4 as a *GBA1*-Associated Protein in CSF and iPS Cells from Parkinson’s Disease Patients with *GBA1* Mutations

**DOI:** 10.3390/ijms25010683

**Published:** 2024-01-04

**Authors:** Rika Kojima, Wojciech Paslawski, Guochang Lyu, Ernest Arenas, Xiaoqun Zhang, Per Svenningsson

**Affiliations:** 1Department of Clinical Neuroscience, Karolinska Institutet, 171 76 Stockholm, Sweden; rika.kojima@ki.se (R.K.);; 2Department of Medical Biochemistry and Biophysics, Karolinska Institutet, 171 77 Stockholm, Sweden

**Keywords:** Parkinson’s disease, *GBA1*, secretome, cerebrospinal fluid, human-induced pluripotent stem cells (hiPSCs), FKBP4

## Abstract

Mutations in the *GBA1* gene increase the risk of developing Parkinson’s disease (PD). However, most carriers of *GBA1* mutations do not develop PD throughout their lives. The mechanisms of how *GBA1* mutations contribute to PD pathogenesis remain unclear. Cerebrospinal fluid (CSF) is used for detecting pathological conditions of diseases, providing insights into the molecular mechanisms underlying neurodegenerative disorders. In this study, we utilized the proximity extension assay to examine the levels of metabolism-linked protein in the CSF from 17 PD patients carrying *GBA1* mutations (GBA1-PD) and 17 idiopathic PD (iPD). The analysis of CSF secretome in GBA1-PD identified 11 significantly altered proteins, namely FKBP4, THOP1, GLRX, TXNDC5, GAL, SEMA3F, CRKL, APLP1, LRP11, CD164, and NPTXR. To investigate *GBA1*-associated CSF changes attributed to specific neuronal subtypes responsible for PD, we analyzed the cell culture supernatant from GBA1-PD-induced pluripotent stem cell (iPSC)-derived midbrain dopaminergic (mDA) neurons. The secretome analysis of GBA1-PD iPSC-derived mDA neurons revealed that five differently regulated proteins overlapped with those identified in the CSF analysis: FKBP4, THOP1, GLRX, GAL, and CRKL. Reduced intracellular level of the top hit, FKPB4, was confirmed via Western Blot. In conclusion, our findings identify significantly altered CSF GBA1-PD-associated proteins with FKPB4 being firmly attributed to mDA neurons.

## 1. Introduction

Mutations in the *GBA1* gene are one of the most common risk factors for Parkinson’s disease (PD) besides *SNCA*, *LRRK2*, and *MAPT* [1,2]. The *GBA1* gene encodes a lysosomal enzyme glucocerebrosidase (GCase), which degrades glucosylceramide into glucose and ceramide. Loss of GCase activity caused by homozygous mutations in the *GBA1* gene leads to Gaucher’s disease (GD) [3]. To date, more than 300 variants in the *GBA1* gene have been identified as pathogenic [4,5,6]. Several *GBA1* mutations are closely related to the clinical subtypes of GD. For example, c.1448T > C (p.Leu483Pro; L483P) is a severe mutation associated with neuronopathic GD type 2 and 3, while c.1226A > G (p.Asn409Ser; N409S) is a mild mutation which mainly accounts for non-neuronopathic GD type 1 [7]. Multicenter analysis of PD patients revealed that carriers of *GBA1* mutations face a several-fold higher risk of developing PD [8]. Among PD-associated *GBA1* variants, c.1093G > C (p.Glu365Lys; E365K) is a common risk mutation in European populations, which increases PD risk when combined with another mutation [9,10]. The E365K mutation affects GCase activity to a much lesser extent compared to other *GBA1* variants like L483P and N409S, and it does not cause GD when mutated biallelically [11,12]. The association between reduced GCase enzyme activity and the pathogenesis of PD remains unclear. However, accounting for the fact that most GD patients do not develop PD despite the severe loss of GCase activities [13] and the discrepancy between residual GCase activities and PD risks, other factors on top of compromised GCase activity are likely involved [14,15].

Cerebrospinal fluid (CSF) is a bodily fluid circulating through the extracellular region of the central nervous system (CNS). CSF serves as both a physical shock absorber for the brain and a regulator of brain homeostasis by removing waste and toxic metabolites from the extracellular space of the brain [16]. The analysis of CSF is employed to evaluate pathological conditions in various diseases. For instance, established CSF biomarkers such as amyloid-β 42, total tau protein, and phosphorylated tau protein are used in the diagnosis of Alzheimer’s disease (AD) [17]. Although reliable CSF biomarkers for PD are yet to be identified, previous reports suggest potential markers including α-synuclein, neurotransmitters, neurofilament light chain, stress markers, and inflammatory markers [18,19]. Also, our recent work identified midkine and 3,4-dihydroxyphenylalanine decarboxylase as useful biomarkers of PD pathology [20]. CSF abnormalities may reflect pathological changes in the CNS of PD. Investigating the impact of *GBA1* mutations on the CSF proteome provides insights into the *GBA1*-associated PD (GBA1-PD) specific molecular basis compared to non-*GBA1*-related PD.

Induced pluripotent stem cell (iPSC)-derived neurons serve as crucial in vitro models for investigating the molecular mechanisms of PD pathophysiology [21]. Advances in iPSC technologies not only offer potential therapeutic alternatives for PD but also allow the study of human neurons in vitro, which was previously challenging. Using established directed differentiation methods, midbrain dopaminergic (mDA) neurons, the most affected cell type in PD, can be generated from iPSCs [22]. The increasing number of studies take advantage of PD patient-derived iPSCs to elucidate the connections between *GBA1* mutations and PD [23,24,25,26,27]. Detailed investigations into the roles of specific gene mutations in PD pathogenesis are achieved through mutation-corrected isogenic control. This approach maintains the same genetic background, except for the mutation of interest, facilitating a thorough understanding of the mutation’s impact [28].

In the current study, we employed the proximity extension assay (PEA), an established tool for CSF screenings [20,29,30], to investigate GBA1-PD-specific secretome alterations in CSF. Our analysis identified significant changes in metabolism-associated proteins in the CSF of GBA1-PD. Moreover, we verified that the CSF secretome alterations were partially recapitulated in the GBA1-PD iPSC-derived mDA neurons. Notably, FK506-binding protein 4 (FKBP4), one of the most significantly altered proteins found in both CSF and supernatant, exhibited increased levels in GBA1-PD iPSC-derived mDA neurons. This comparison between CSF and mDA neuron culture supernatant allowed us to successfully identify specific secretome changes in GBA1-PD attributed to the mDA neuronal population.

## 2. Results

### 2.1. Proteins Significantly Altered in GBA1-PD CSF

To characterize secretomic signatures of GBA1-PD CSF, we analyzed CSF proteins of 17 GBA1-associated PD (GBA1-PD) patients and 17 age- and sex-matched idiopathic PD (iPD) patients. Given the significance of GCase in lipid metabolisms, we opted to include proteins from the Olink Metabolism 96 panel for our analysis. Out of these, a total of 53 proteins with sufficient detection levels proceeded to the subsequent analysis. The list of proteins included in the analysis is found in Appendix A. Group fold changes in normalized protein expression (NPX) values between GBA1-PD and iPD and Benjamini–Krieger–Yekutieli adjusted *p*-values (q-values) were used to construct volcano plots highlighting significant markers (Figure 1a, q < 0.05). Eleven significantly altered proteins were identified, marked red in a forest plot (Figure 1b), showing log_2_ fold changes with the standard error of the difference (SE). The significantly altered proteins were FK506-binding protein 4 (FKBP4), Thimet oligopeptidase 1 (THOP1), Glutaredoxin-1 (GLRX), Thioredoxin domain containing 5 (TXNDC5), Galanin (GAL), Semaphorin 3F (SEMA3F), CT10 regulator of kinase-like protein (CRKL), Amyloid-like protein 1 (APLP1), low-density lipoprotein receptor-related protein 11 (LRP11), Sialomucin core protein 24 (CD164), and neuronal pentraxin receptor (NPTXR) (Figure 1c). Twenty-one proteins were identified as the hit candidate proteins with a *p*-value < 0.05 without correcting for multiple comparisons (Figure 1b).

### 2.2. CSF CRKL Levels and UPDRS-III Scores Oppositely Correlate in GBA1-PD and iPD

To investigate whether there is any association between the CSF hit candidate protein levels and clinical scores, a correlation analysis between patient demographics and NPX values was performed. Demographic characteristics of patients are summarized in Table 1. Several proteins in iPD and GBA1-PD CSF significantly correlated with clinical scores with a *p*-value < 0.05 (Appendix A). Among them, CRKL, one of the significantly upregulated proteins in GBA1-PD CSF, positively correlated with the Unified Parkinson’s Disease Rating Scale part III (UPDRS-III) scores in iPD, which showed the opposite correlation in GBA1-PD (Appendix A).

### 2.3. Secretome Changes in GBA1-PD iPSC-Derived mDA Neuron Culture Supernatant

Next, to validate if the observed secretomic changes were reproduced in vitro, we investigated cell culture supernatant of iPSC-derived mDA neurons, which were generated from *GBA1* N409S heterozygous PD patient-derived iPSCs and their isogenic control iPSCs. Genotyping analysis confirmed the successful correction of the heterozygous *GBA1* N409S mutation in the isogenic iPSC line (Appendix A). Immunostaining validated the expression of pluripotency markers (octamer-binding transcription factor 4 (OCT4), TRA-1-60, NANOG) in both the *GBA1* N409S mutant and isogenic iPSCs (Appendix A). Both *GBA1* N409S mutant and isogenic iPSCs were successfully directed to midbrain floor plate (mFP) progenitors expressing forkhead box protein A2 (FOXA2), engrailed 1 (EN1), and LIM homeobox transcription factor 1 alpha (LMX1A) with comparable efficiencies (Appendix A). mFP progenitors were further differentiated into mature mDA neurons expressing tyrosine hydroxylase (TH), nuclear receptor-related 1 protein (NURR1), and microtubule-associated protein 2 (MAP2) (Figure 2a). Enzyme activity assay confirmed that day 60 GBA1-PD mDA neurons retained about 50% of GCase activities of isogenic control neurons, which agrees with previous reports on the residual activities of *GBA1* N409S heterozygous neurons [31,32] (Figure 2b). Overnight cell culture supernatant was collected on day 60 of differentiation and analyzed with the Olink Metabolism 96 panel. Due to the low number of samples and the exploratory purpose of the investigation, the *p*-values were not adjusted for multiple comparisons. Sixteen proteins with a *p*-value < 0.05 were identified as the hit candidate proteins for supernatant analysis (Figure 2c). Among them, five proteins overlapped with the significantly altered proteins in CSF analysis, namely CRKL, GAL, FKBP4, GLRX, and THOP1 (Figure 2d).

### 2.4. Secretome Analysis on GBA1-PD iPSC-Derived mDA Neurons Highlights the GBA1-PD CSF Secretome Alterations Attributed to the mDA Neuronal Population

To compare the secretomes of CSF and supernatant, we focused on eight proteins identified in both analyses. The NPX changes in these eight proteins exhibited similar trends in both CSF and supernatant, with the exception of GAL and FAM3 metabolism-regulating signaling molecule C (FAM3C) (Figure 3a). Among the significantly altered proteins in CSF, five were also found in the supernatant from GBA1-PD iPSC-derived mDA neurons, of which FKBP4, THOP1, GLRX, and CRKL were upregulated in GBA1-PD (Figure 3b). To validate the findings, we used the Western Blot method to assess the intracellular protein levels of FKBP4, the most significantly altered protein in the CSF analysis, on D60 mDA neurons (Figure 3c). Normalized FKBP4 expression was significantly increased in GBA1-PD iPSC-derived mDA neurons compared with isogenic control neurons, supporting the observed alterations in FKBP4 levels in GBA1-PD mDA neuron culture supernatant (Figure 3d).

## 3. Discussion

This study represents the first attempt to compare secretome signatures of CSF from PD patients with cell culture supernatant from PD-iPSC-derived mDA neurons and isogenic controls. As highlighted by Tüshaus et al., CSF contains a variety of proteins secreted from diverse cell types in the central and peripheral nervous systems, which makes it challenging to determine the cellular origin of dysregulated CSF proteins [33]. To address this, we conducted secretome analysis on both CSF and the supernatant of iPSC-derived mDA neurons, clarifying the contributions of specific cell types in the GBA1-PD brain secretomes. Our study successfully identified mDA neuron-attributed CSF alterations, introducing a novel strategy for examining CSF secretome. The five identified mDA neuron-attributed CSF proteins offer valuable insights into the *GBA1*-specific secretome alterations, contributing to the understanding of molecular mechanisms underlying the pathogenesis of GBA1-PD.

FKBP4 is an immunophilin protein that binds to immunosuppressants like FK506 and rapamycin [34,35]. FKBP4 is also known as a co-chaperone of heat shock protein 90, involving the regulation of steroid receptor activity [36,37]. Several studies suggest a connection between FKBP4 and PD. It has been reported that FKBP4 acts as a modulator of the α-synuclein-evoked immune response [38] and inhibition of FKBP4 reduces α-synuclein aggregations [39]. Fusco et al. identified a compound heterozygous mutation in RET proto-oncogene and the *FKBP4* genes in early-onset PD which disrupts the RET51/FKBP4 complex [40]. Proteome analysis of the locus coeruleus in PD patients revealed downregulated levels of FKBP4 [41]. Kim et al. highlighted the protective effect of melatonin-mediated downregulation of FKBP4 on neuronal mitochondria dysfunctions [42]. While FKBP4 is recognized for its involvement in diverse cellular processes beyond immunosuppression, further exploration is essential to understand its role in PD pathology.

THOP1 is an oligopeptidase that hydrolyses small neuropeptides shorter than 17 amino acids in length [43]. THOP1 is discussed in its roles in AD, particularly its protective effect. It is known that THOP1 degrades amyloid-β precursors [44]. Overexpression of THOP1 has a neuroprotective effect on amyloid-β toxicity [45]. A recent study reported that THOP1 levels were significantly increased in CSF of AD patients, implicating the possibility of THOP1 as a biomarker for AD [46,47].

GLRX is a thioltransferase regulating redox homeostasis by restoring S-glutathionylated proteins to the reduced state [48]. Upregulation of GLRX expression is considered to be neuroprotective, while it is associated with inflammatory responses in microglial cells [49]. In terms of the association with PD, loss of GLRX causes the accumulation of S-glutathionylation proteins, which triggers apoptosis in DA neurons [50,51]. This is confirmed by a study that showed decreased GLRX protein levels in post-mortem PD brains, particularly within DA neurons [52]. Also, we have observed decreased levels of GLRX in PD patients’ CSF compared to healthy control using the PEA assay before [20]. On the other hand, it was found that *GLRX* expression is upregulated in SNCA-A53T iPSC-derived DA neurons under rotenone-induced oxidative stress conditions [53], suggesting a compensatory response in the impaired but surviving neurons.

CRKL is an adaptor protein involved in signal transduction and mediating numerous biological processes. CRKL protein contains Sarcoma Homology 2 and Sarcoma Homology 3 domains, which regulate protein–protein interactions activated by protein tyrosine kinases [54,55]. The *CRKL* gene is also known as an oncogene upregulated in cancer cells, specifically associated with lung and gastric cancer [56,57]. In our previous study, we observed lower levels of CSF CRKL in PD patients compared to controls [20]. Although the roles of CRKL in PD pathology have not been specified yet, CRKL was recently found specifically expressed in a subtype of adult human mDA neurons [58] and is known to play a key role in mDA neuron development by acting downstream of the Reelin signaling pathway [59,60,61]. We showed an association between CSF CRKL levels and UPDRS-III scores, which suggests a yet-to-known role of CRKL in motor function. Given that CRKL levels correlate in opposite directions for GBA1-PD and iPD, CRKL could receive genotype-specific regulation in the brain.

GAL, a neuroendocrine peptide distributed in the central and peripheral nervous system [62,63], exerts various functions through its G protein-coupled receptors GALR1, GALR2, and GALR3 [64]. GAL expression increases after nerve injury, suggesting a neuroprotective role in promoting survival [65,66]. The neuroprotective potential of GAL has gathered attention as a potent therapeutic target for AD [67], attributed to its effectiveness against AD-related amyloid-β toxicity [68,69]. On the other hand, upregulation of GAL in AD is considered detrimental, as it inhibits acetylcholine release from cholinergic neurons [70], leading to cognitive impairment resulting from cholinergic system dysfunction [71]. While the functions of GAL in PD were less explored compared to AD, existing studies suggest that GAL inhibits the activities of DA neurons [72], and GAL injection to the ventral tegmental area induces depression-like behavior in mice [73]. In our study, we observed increased GAL levels in the GBA1-PD CSF, contrasting with findings from the GBA1-PD iPSC-derived mDA neuron supernatant. The inconsistency between CSF and supernatant analysis may be explained by the unique regulation of GAL expression in different brain regions and cell types [74]. In a DA neuron-dominant environment, GAL levels could yield different outcomes than CSF. Given GAL’s role in inhibiting DA neuron activities, it is reasonable to expect downregulated GAL levels in affected neurons, potentially compensating for the degenerated dopaminergic function caused by GCase deficiency. It is worth noting that GAL levels in CSF of AD patients were not altered [75,76].

A few studies investigated the CSF alterations by focusing on GBA1-PD previously. Kaiser et al. identified six differentially expressed proteins in the CSF of GBA1-PD patients compared to unaffected *GBA1* mutation careers, which indicates increased neuroinflammatory events linked to the loss of dopaminergic neurons in GBA1-PD [77]. In our findings, the immune-related protein FKBP4 was identified, emphasizing a potential association of GBA1 with the immune system. On the other hand, Lerche et al. explored inflammatory markers in CSF from PD_GBA_WT_ and PD_GBA_ patients, which revealed similar levels of inflammatory markers irrespective of the *GBA1* mutation status [78]. Further investigations are necessary to clarify whether the proposed immunological alterations are specific to GBA1-PD or represent a general phenomenon in PD.

While we employed a robust, well-established differentiation protocol that provides high-quality mDA neurons with good reproducibility, it is important to acknowledge the presence of other neuronal subtypes in the culture, which could influence the analysis outcomes. Consequently, the data should be interpreted with consideration to the potential contributions from these minor subtypes. Also, we did not examine other cells with the same mutation but with different genetic backgrounds. Since other *GBA1* N409S heterozygous iPSCs and isogenic control are not available, we compared the iPSCs to CSF from PD patients, which explores a much greater diversity of genetic backgrounds. We believe that the identification of common hits in these two very different samples provides strong support for the results hereby described. It must be noted that our cohorts predominantly consisted of women, which could potentially influence the results. However, evaluating the impact of the sex bias and drawing definitive conclusions is challenging given the relatively small size of our cohorts. More studies in larger replication cohorts will be needed to address the importance of observed changes. Additionally, further investigations on FKBP4 and other hits are necessary to understand the roles of the identified proteins in the pathogenesis of GBA1-PD. Such studies may not only consider proteins detected in CSF from GBA-PD patients and in the supernatant of GBA-PD iPSC-derived mDA neurons but also proteins only found in the CSF secretome representing glia or non-dopamine neuronal changes in GBA-PD.

## 4. Materials and Methods

### 4.1. Chemicals

Unless otherwise stated, all chemicals were purchased from Sigma-Aldrich (Merck KGaA, Darmstadt, Germany) and were of analytical grade. All solutions were made using Milli-Q deionized water (Merck).

### 4.2. Patients with Parkinson’s Disease

The patients were followed by a movement disorder specialist and fulfilled the clinical diagnosis criteria for PD [79]. Patients with any other serious neurological or psychiatric disorders or cancer were excluded from the cohorts. CSF samples from patients were collected by a movement disorder specialist from individuals who underwent lumbar punctures in neurological clinics within Region Stockholm, and where excess collected sample volume had been stored. The study individuals gave written consent to the storage of their samples for future use in studies, and the study has been approved by the Swedish Ethical Review Authority (dnr 2020-03684). The PD patients included were additionally part of the Stockholm Biopark cohort (dnr 2019-04967) [80] and were clinically assessed by a movement disorder specialist. The disease severity of the PD patients was evaluated with the Unified Parkinson’s Disease Rating Scale part III (UPDRS-III) [81] and Hoehn and Yahr scale (HY), and their cognition using the Montreal Cognitive Assessment (MoCA) [82] scale. PD medications are summarized as L-dopa equivalent doses (LEDD) [83]. The *GBA1* genotyping was performed with Sanger sequencing. Demographic characteristics are summarized in Table 1.

### 4.3. CSF Collection

The standardized lumbar puncture was performed as described before [84,85]. Briefly, CSF collection was performed sitting up at the L3-L4 interspace, in accordance with the Alzheimer’s Disease Neuroimaging Initiative recommended protocol. Samples were collected into sterile polypropylene tubes; the first 2 mL were discarded and between 10 and 12 mL CSF from the first portion was collected and gently mixed to minimize the gradient influence. Cell counts were measured, and samples were centrifuged in the original tube at 4000 rpm for 10 min at 4 °C. CSF samples were aliquoted, frozen on dry ice, and stored at −80 °C until assay. The time between sample collection and freezing was a maximum of 30 min.

### 4.4. Cell Lines

*GBA1* N409S heterozygous PD patient-derived iPSCs (NH50187, MUTANT) and their isogenic control cells (NH50186, ISOGENIC) were obtained from the Rutgers University Cell and DNA Repository (RUCDR). Normal karyotypes of NH50187 and NH50186 were confirmed by RUCDR. Genotypes, pluripotency, and mycoplasma infection were routinely assessed before starting new experiments (Appendix A). Undifferentiated iPSCs were grown on Laminin 521 (Biolamina, Stockholm, Sweden, LN521)-coated plates and maintained in Essential 8 Flex media (Thermo Fisher Scientific, Waltham, MA, USA, A2858501) at 37 °C, 5% CO_2_ until starting differentiation.

### 4.5. Midbrain Dopaminergic Neuron Differentiation

NH50187 and NH50186 were directed-differentiated into mDA neurons based on the previously reported protocols with some modifications [86,87]. Briefly, iPSCs were plated at a density of 200,000 cells/cm^2^ on Geltrex (Thermo Fisher Scientific, A1413201) +Laminin 511 (Biolamina, LN511)-coated plates and cultivated in Neurobasal/N2/B27 (Life Technologies, Carlsbad, CA, USA, 21103049) medium with 2 mM L-glutamine (Invitrogen, Carlsbad, CA, USA, 25030081) through the entire protocol. Supplements: Day 0–10, N2 supplement (Thermo Fisher Scientific, 17502001); day 0–7, 250 nM LDN193189 (Fisher Scientific, Waltham, MA, USA, 17138019), 10 µM SB431542 (Tocris, Bristol, UK, 1614), and 1 µM Purmorphamine (Tocris, 4551). CHIR99021 (Tocris, 4423) was added at 0.7 µM day 0–4, 7.5 µM day 4–10, and 3 µM day 10–11. Day 10–60, cells cultured in maturation media consisting of 20 ng/mL brain-derived neurotrophic factor (BDNF, R&D Systems, NE Minneapolis, MN, USA, 248-BDB-050), 20 ng/mL glial cell line-derived neurotrophic factor (GDNF, R&D Systems, 212-GD-050), 1 ng/mL transforming growth factor type β3 (TGFβ3, R&D Systems, 243-B3-010), 200 µM ascorbic acid (Sigma-Aldrich, Burlington, MA, USA, A4403), 200 µM Dibutyryl cAMP (dbcAMP, Sigma-Aldrich, D0627), and 3 µM CHIR99021. On day 11, cells were dissociated into single cells and were replated onto plates coated with polyornithine (Sigma-Aldrich, P4957) and Laminin 511 at a density of 800,000 cells/cm^2^. A total of 10 µM DAPT (Sigma-Aldrich, 2634) was added from day 12. A cocktail of small molecules consisting of 10 μM GW3965 (Sigma-Aldrich, G6295), 1 μM PD0325901 (Sigma-Aldrich, PZ0162), and 5 μM SU5402 (Sigma-Aldrich, SML0443) was administered from day 12 to day 15. On day 16, cells were replated as on day 11 and cultured until day 60 in maturation media. Cell culture media were replaced daily until day 22 and every other day from day 23. A total of 10 µM Y27632 (Tocris, 1254) was supplemented for 24 h after each replating. Penicillin + Streptomycin (Thermo Fisher Scientific, 15140122) was supplemented from day 23.

### 4.6. Conditioned Cell Culture Media Collection

Day 59 mDA neurons were incubated in 1 mL/well of Neurobasal minus phenol red (Thermo Fisher Scientific, 12348017) +B27 +L-Glutamine supplemented with 20 ng/mL BDNF, 20 ng/mL GDNF, 1 ng/mL TGFβ3, 200 µM ascorbic acid, 200 µM dbcAMP, and 10 µM DAPT for 24 h. On day 60, cell culture supernatant was collected in a 1.5 mL Eppendorf and snap froze with dry ice. Collected samples were stored at −80 °C until use. Supernatant from two individual wells per cell line was collected for three independent differentiations.

### 4.7. Protein Quantification

Following the cell culture media sampling, neurons were harvested for protein quantification. Cells were lysed with RIPA buffer (Thermo Fisher Scientific, 89901) containing proteinase inhibitor cocktail (Sigma-Aldrich, 04693159001) and phosphatase inhibitor cocktail (Roche, Basel, Switzerland, 04906837001), incubated on ice for 30 min, and then lysed with sonication for 3 × 5 s at 30% amplitude. Cell lysates were spun down at 16,000× *g* for 20 min at 4 °C. Collected supernatants were stored at −20 °C until use or immediately subjected to protein quantification with the Pierce BCA Protein Assay Kit (Thermo Fisher Scientific, 23225). BCA analysis was carried out by following the manufacturer’s instructions.

### 4.8. Western Blot

Cell lysates were diluted and mixed with 4x SDS-loading buffer (Bio-Rad, Hercules, CA, USA, 1610747) to the final concentration of 1 mg/mL total protein. A total of 10 µg of the total protein was loaded into each well. Semi-dry transfer was performed at 2.5 A for 11 min onto the PVDF membrane (Bio-Rad, 1704275). Membranes were fixed with 4% paraformaldehyde (PFA) supplemented with 0.02% glutaraldehyde for 30 min and washed with water three times before blocking. Membranes were blocked in Intercept (TBS) blocking buffer (LI-COR Biosciences, Lincoln, NE, USA, 927-60001) for 1 h at room temperature. Primary antibodies were applied overnight at 4 °C. The next day, membranes were washed three times with TBS-T and incubated in the secondary antibodies for 1 h at room temperature. The membranes were visualized on LI-COR Odyssey CLx. Densitometric analysis was performed on the Image Studio software version 5.2.5. Signal intensities from the objected bands were normalized to that of the housekeeping protein. The following primary and secondary antibodies were used: anti-GAPDH (6C5) (Santa Cruz Biotechnology, Dallas, TX, USA, sc-32233) 1:1000, anti-FKBP4 (Proteintech, Manchester, UK, 10655-1-AP) 1:1000, IRDye 800CW goat anti-rabbit IgG (LI-COR Biosciences, 926-32211), and IRDye 680RD goat anti-mouse IgG (LI-COR Biosciences, 926-68070).

### 4.9. Immunocytochemistry

For mDA neuron markers, day 16 mDA neuron progenitors were seeded at around 300,000 cells/well density in a 96-well plate. Cells were cultured until day 28 or 60, as described above. On day 28 or 60, cells were fixed with 4% PFA for 20 min at 4 °C and blocked with 5% Donkey serum (Sigma-Aldrich, D9663) in 0.02% PBS-Triton X-100 for 1 h at room temperature. For pluripotency markers, undifferentiated iPSCs were seeded in a 96-well plate, cultured until achieving 80% confluence, then fixed as described and subjected to immunostaining. For floor plate markers, day 11 mFP progenitors were seeded at around 300,000 cells/well density in a 96-well plate. On the next day, cells were fixed as described and subjected to immunostaining. Fixed cells were incubated in the primary antibody overnight at 4 °C. The following primary antibodies were used: anti-TH (Pel-Freez, Rogers, AR, USA, P40101-150) 1:1000, anti-NURR1 (R&D Systems, PPN140400) 1:1000, anti-MAP2 (Abcam, Cambridge, UK, ab5392) 1:1000, anti-FOXA2 (R&D Systems, AF2400) 1:1000, anti-LMX1A (Merck KGaA, AB10533) 1:1000, anti-EN1 (DSHB, Iowa city, IA, USA, 4611-c) 1:50, anti-OCT4 (R&D Systems, AF1759) 1:200, anti-TRA-1-60 (Invitrogen, MA1-023) 1:500, and anti-Nanog (Cell Signaling Technology, Danvers, MA, USA, 4903S) 1:200. Next day, cells were washed with DPBS three times and incubated in the secondary antibody for 1 h at room temperature. The following secondary antibodies were used: donkey anti-rabbit Alexa Fluor 555 (Thermo Fisher Scientific, A32794) 1:800, donkey anti-mouse Alexa Fluor 647 (Thermo Fisher Scientific, A31571) 1:800, donkey anti-mouse Alexa Fluor 488 (Thermo Fisher Scientific, A32766) 1:800, goat anti-chicken Alexa Fluor 488 (Thermo Fisher Scientific, A11039) 1:800, donkey anti-goat Alexa Fluor 647 (Thermo Fisher Scientific, A21447) 1:800, and DAPI (Sigma-Aldrich, D9542) 1:1000. Cells were visualized using Zeiss LSM 880 or 900 (Carl Zeiss AG, Oberkochen, Germany).

### 4.10. Genotyping of iPSCs

Genomic DNA of undifferentiated *GBA1* N409S mutant/isogenic iPSCs was extracted using the DNeasy blood and tissue kit (Qiagen, Venlo, The Netherlands, 69504) by following the manufacturer’s instructions. DNA fragments containing c.1226A > G (p.Asn409Ser; N409S) were amplified via PCR with forward primer 5′-AGC TAG CCT GCC CTT TTG AG-3′ and reverse primer 5′-GCT TTT CTG CAT CGC AGT CC-3′. PCR products were isolated by using gel electrophoresis and purified by using the QIAquick Gel Extraction kit (Qiagen, 28704). Obtained PCR amplicons were sequenced via Sanger DNA sequencing.

### 4.11. GCase Activity Assay

Day 60 mDA neurons were harvested with TNT buffer (100 mM Tris–HCl (pH 7.4), 100 mM NaCl, 0.2% Triton X-100) containing proteinase inhibitors as described above. A total of 10 μg/well of the total protein was mixed with the activity assay buffer (Citrate–Phosphate buffer pH 5.4, 1% bovine serum albumin, 0.25% Triton X-100, 0.25% mM Sodium taurocholate, 0.1% EDTA) and loaded into a black 96-well plate in duplicate. A total of 1 mM conduritol-β-epoxide (CBE) was added to the control wells. The reaction was initiated by the addition of 1 mM 4-methylumbelliferyl β-d-glucopyranoside (Glycosynth, Warrington, UK, 44059). Fluorescence was measured on a Tecan Spark 10 M (Ex: 360/Em: 449) every 10 min for 6 h at 37 °C. Signals from CBE-treated controls were subtracted as background. GCase activity was represented as a slope calculated through linear regression of fluorescence intensity from 100 to 300 min (21 data points). The linear regression R^2^ was within 0.9877 to 0.9997.

### 4.12. Proximity Extension Assay (PEA)

Proteins in CSF and cell culture media were analyzed by Olink Proteomics PEA technology to detect multiple proteins at high specificity and sensitivity. For GBA1-PD CSF, 12 out of 17 samples were previously analyzed with the Olink Metabolism 96 panel (Cohort I). The additional five samples were newly analyzed with the Olink Metabolism 96 panel (Cohort II). To combine the data from two cohorts, we normalized all GBA1-PD CSF data using bridging samples. We confirmed that the data from Cohort II fit with Cohort I after normalization, and the whole dataset was distributed normally (Appendix A). The list of pre-selected proteins included in the panel, and Olink panel validation data are freely available online (https://www.olink.com/data-you-can-trust/validation/, accessed on 8 December 2023). Protein expression is represented as Normalized Protein eXpression (NPX) values, a relative protein quantification unit transformed as a log_2_ scale. Proteins were excluded where >30% fell under the limit of detection (LOD). A total of 53 proteins from the Metabolism 96 panel were included in the downstream analysis (Appendix A). The CSF protein levels were normalized to the total protein content for each sample. The cell culture supernatant protein levels were normalized to the total protein content for each cell lysate.

### 4.13. Statistics

GraphPad Prism v9.00 was used for the generation of all graphs and statistical analysis unless stated otherwise. Error bars represent the standard error of the mean (SEM). NPX values from Cohort II were normalized using bridging samples of each group and confirmed to fit with Cohort I after the normalization. After normalization, outliers were identified by the robust regression and outlier removal (ROUT) method (Q = 1%) and removed from the subsequent analysis. Demographics were analyzed with the Mann–Whitney U test. For CSF analysis, multiple two-tailed Student’s *t*-tests with Benjamini, Krieger, and Yekutieli’s false discovery rate (FDR) correction were used for q-value calculation. The FDR cutoff of <0.05 was used for the determination of the significantly altered proteins. The Spearman’s rank correlation coefficient was used for correlation analysis; *p*-values for Spearman’s rank correlation coefficient were not adjusted for multiple comparisons. For cell culture supernatant analysis, multiple two-tailed Student’s *t*-test without correction for multiple comparisons was used. Proteins with a *p*-value < 0.05 were defined as the hit candidate proteins. Log_2_(Fold Change), *p*-values, and q-values of the significantly altered CSF proteins (q < 0.05) and the hit candidate iPSC proteins (*p* < 0.05) were listed in Appendix A. Significance was set as follows: * = *p* < 0.05; ** = *p* < 0.01; *** = *p* < 0.001.

## 5. Conclusions

Our study unveiled proteome alterations in the CSF of GBA1-PD associated with the mDA neuronal population. Initially, we identified significantly altered proteins in GBA1-PD CSF using the PEA platform. To assess the contribution of mDA neurons to these CSF alterations, we differentiated GBA1-PD patient-derived iPSCs into mDA neurons and analyzed their secretomes. Among the significantly altered proteins in CSF, FKBP4, THOP1, GLRX, GAL, and CRKL also exhibited differential regulation in the secretome of GBA1-PD iPSC-derived mDA neurons. Finally, we confirmed the upregulation of FKBP4 protein levels in GBA1-PD iPSC-derived mDA neurons. In summary, our findings not only identify potential contributing proteins in GBA1-PD pathogenesis attributed to mDA neurons but also underscore the potential of iPSC-derived neurons as a valuable tool for exploring and validating the molecular mechanisms of the CSF secretome changes in vitro.

## Figures and Tables

**Figure 1 ijms-25-00683-f001:**
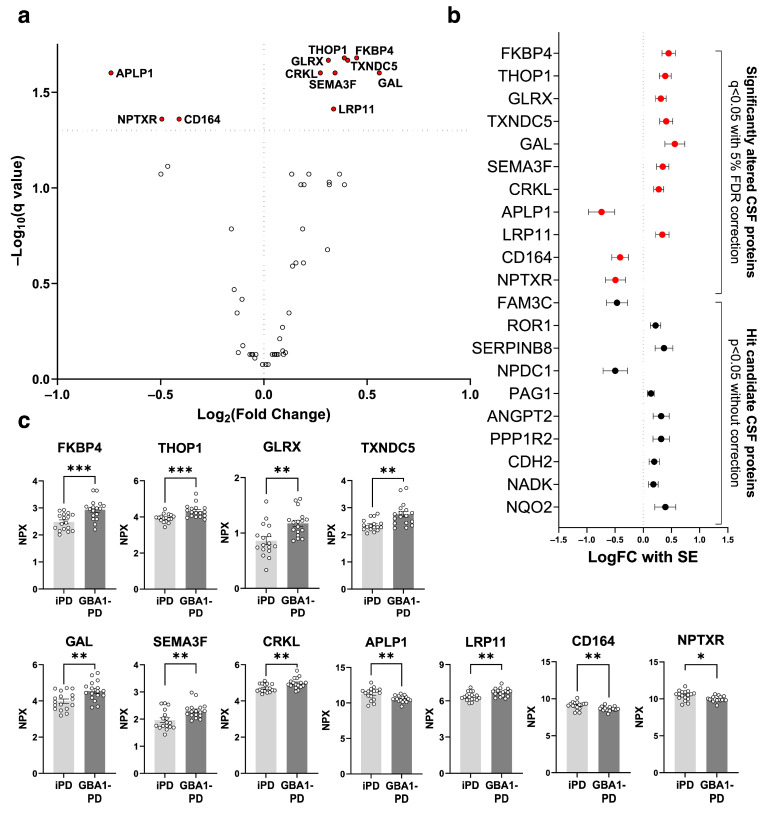
Secretome analysis on *GBA1*-associated Parkinson’s disease (GBA1-PD) and idiopathic PD (iPD) cerebrospinal fluid (CSF). (**a**) Volcano plot showing log_2_ fold change and −log_10_ 5% false discovery rate (FDR)-adjusted *p*-value (q-value) in analyzed protein levels between GBA1-PD and iPD CSF (n = 17). The significantly altered CSF proteins are indicated in red. (**b**) Forest plot displaying log_2_ fold change with the standard error of the difference (LogFC with SE) of the hit candidate proteins in the CSF analysis ordered in ascending *p*-values. The significantly altered CSF proteins are indicated in red and the hit candidate CSF proteins are indicated in black. (**c**) Bar graphs of the significantly altered proteins. Levels representing the normalized protein expression (NPX) (n = 17). The light bar indicates iPD and the dark bar indicates GBA1-PD. Data are presented as the mean  ±  SEM. Two-tailed Student’s *t*-test with Welch’s correction, * = *p* < 0.05, ** = *p* < 0.01, *** = *p* < 0.001.

**Figure 2 ijms-25-00683-f002:**
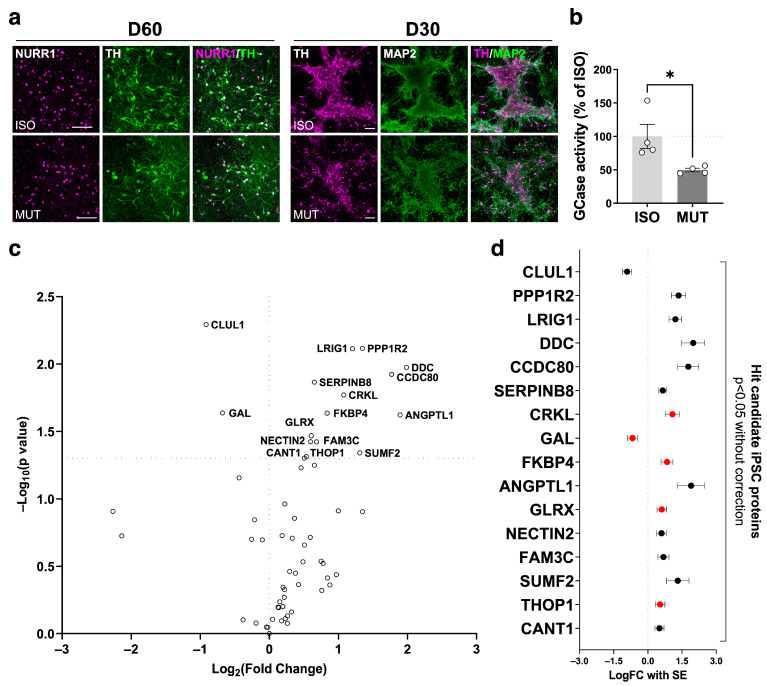
Secretome analysis on GBA1-PD-induced pluripotent stem cell (iPSC)-derived midbrain dopaminergic (mDA) neuron culture supernatant. (**a**) Representative immunostaining of mDA markers (NURR1, TH, and MAP2) on mDA neurons differentiated from *GBA1* N409S mutant (MUT) or isogenic (ISO) iPSCs at day 60 (left) and 30 (right). Scale bars, 100 µm. (**b**) GCase activity assay on day 60 *GBA1* N409S mutant and isogenic iPSC-derived mDA neurons (n = 4). (**c**) Volcano plot showing log_2_ fold change and −log_10_ *p*-value in analyzed protein levels between *GBA1* mutant and isogenic iPSC-derived mDA neuron culture supernatant (n = 6). (**d**) Forest plot displaying log_2_ fold change with SE of the proteins with a *p*-value < 0.05 in the supernatant analysis ordered in ascending *p*-values. Hit candidate iPSC proteins in common with the significantly altered CSF proteins are indicated in red. Hits found exclusively in the supernatant are indicated in black. Data are presented as the mean ± SEM. Two-tailed Student’s *t*-test, * = *p* < 0.05.

**Figure 3 ijms-25-00683-f003:**
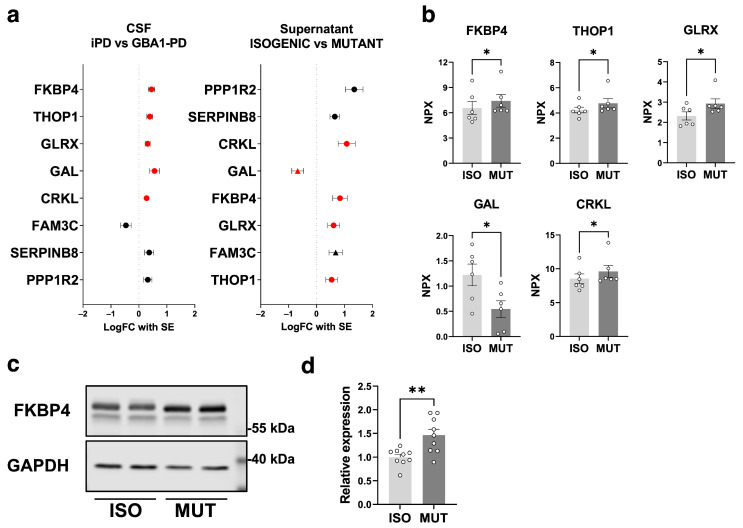
Secretome analysis of the supernatant of GBA1-PD iPSC-derived mDA neurons emphasizes CSF alterations associated with the mDA neuronal populations. (**a**) Forest plots displaying log_2_ fold change with SE of the eight proteins identified both in the CSF analysis (left) and the supernatant analysis (right) ordered in ascending *p*-values. The significantly altered CSF proteins were indicated in red. Hits found exclusively in the CSF or the supernatant are indicated in black. Triangle symbols in the supernatant plot represent the proteins changed in the opposite direction as the CSF analysis. (**b**) Bar graphs illustrating the NPX of five proteins in the supernatant analysis, which are identified in the CSF analysis as significantly altered proteins (n = 6). (**c**) Representative blots showing FKBP4 expression in D60 mDA neurons differentiated from *GBA1* N409S mutant or isogenic iPSCs. (**d**) Bar graphs of densitometric quantification of relative FKBP4 levels (n = 9). Data are presented as the mean ± SEM. Two-tailed Student’s *t*-test, * = *p* < 0.05, ** = *p* < 0.01.

**Table 1 ijms-25-00683-t001:** Analysis of cohort’s characteristics represented as mean (standard deviation) or median (range).

	GBA1-PD	iPD	*p*-Value ^1^
n	17	17	>0.99
Female:Male	5:12	5:12	>0.99
Age mean (SD)	63.29 (9.726) n = 17	63.29 (8.908) N = 17	0.7658
Duration of disease in years	4.0 (0–10) n = 17	2.29 (0.04–18.29) N = 17	0.6768
HY ^2^	2 (1–3) n = 17	2 (1–4) N = 17	0.9562
UPDRS-III ^3^	26 (5–62) n = 15	27.5 (10–54) N = 16	0.4287
MoCA ^4^	26 (9–29) n = 16	26.0 (16–29) N = 16	0.5800
LEDD ^5^	710 (0–1805) n = 17	425.6 (0–1080) N = 17	0.1724

^1^ *p*-value for the Mann–Whitney U test. ^2^ Hoehn and Yahr Scale. ^3^ Unified Parkinson’s Disease Rating Scale part III. ^4^ Montreal Clinical Assessment. ^5^ Levodopa Equivalent Daily Dose.

## Data Availability

The datasets and materials used and/or analyzed during the current study are available from the corresponding author upon reasonable request.

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
