# Peer review of "Secretome Analyses Identify FKBP4 as a GBA1-Associated Protein in CSF and iPS Cells from Parkinson’s Disease Patients with GBA1 Mutations"

_ijms, 2024, doi:10.3390/ijms25010683_

Round 1
Reviewer 1 Report
Comments and Suggestions for Authors
The article is very interesting, the research was conducted and interpreted correctly. The naming of mutations at the DNA and protein levels requires correction (it should be adapted to the latest relevant recommendations). Explanations of abbreviations must be added under the figures. I suggest that the discussion refer to the issue of the predominance of women in both research groups (GBA-PD and iPD), and also write that it is difficult to draw firm conclusions based on the research results of such small groups.
Author Response
Responses to the comments from reviewer 1
The article is very interesting, the research was conducted and interpreted correctly. The naming of mutations at the DNA and protein levels requires correction (it should be adapted to the latest relevant recommendations).
Response: Changed gene names in Italics. Corrected inconsistency in the gene name (GBA→GBA1) and updated nomenclature to E365K, N409S and L483P.
Explanations of abbreviations must be added under the figures.
Response: Indicated abbreviations are now explained in the figure legends.
I suggest that the discussion refer to the issue of the predominance of women in both research groups (GBA-PD and iPD), and also write that it is difficult to draw firm conclusions based on the research results of such small groups.
Response: Text on this point is added to lines 279-282 in the discussion.
Reviewer 2 Report
Comments and Suggestions for Authors
This is a rather straightforward study by experienced groups that work on clinical trials and biomarkers in parkinsonism. Mutations in the GBA1 gene, encodes a lysosomal enzyme glucocerebrosidase (GCase), are one of the most common risk factors for Parkinson’s disease (PD). The association between reduced GCase enzyme activity and the pathogenesis of PD remains unclear. The authors previously identified midkine and 3,4-dihydroxyphenylalanine decarboxylase as useful cerebrospinal fluid (CSF) biomarkers of PD pathology (20). In the current study, they investigated GBA-PD specific secretome alterations on both CSF and the supernatant of iPSC-derived midbrain dopaminergic (mDA) neurons and identified overlapped five significantly altered CSF proteins, FKBP4, THOP1, GLRX, GAL, 21 and CRKL, as most significantly altered proteins.
Based on the above results, they propose that the comparison between CSF and mDA neuron culture supernatant allowed us to successfully identify specific secretome changes in GBA-PD attributed to the mDA neuronal population.
Although the role of FKBP4, THOP1, GLRX, GAL, 21 and CRKL on GBA-PD patients remains unknown, I think this work is well organized and a nice first contribution to the importance of the secretome analysis on both CSF and the supernatant of iPSC-derived mDA neurons of GBA-PD patients.
I have some concerns to address as below.
Major points:
1. I would not think FKBP4 is the most significantly altered protein in the CSF analysis (lane 164), What data did the authors use to make that decision?
2. Why are the six proteins, TXNDC5, SEMA3F, APLP1, LRP11, CD164, and NPTXR, detected in CSF secretome in GBA-PD patient not detected in the supernatant of GBA-PD iPSC-derived mDA neurons? I would think this was due to the difference in sensitivity in both analyses.
Author Response
Responses to the comments from Reviewer II
This is a rather straightforward study by experienced groups that work on clinical trials and biomarkers in parkinsonism. Mutations in the GBA1 gene, encodes a lysosomal enzyme glucocerebrosidase (GCase), are one of the most common risk factors for Parkinson’s disease (PD). The association between reduced GCase enzyme activity and the pathogenesis of PD remains unclear. The authors previously identified midkine and 3,4-dihydroxyphenylalanine decarboxylase as useful cerebrospinal fluid (CSF) biomarkers of PD pathology (20). In the current study, they investigated GBA-PD specific secretome alterations on both CSF and the supernatant of iPSC-derived midbrain dopaminergic (mDA) neurons and identified overlapped five significantly altered CSF proteins, FKBP4, THOP1, GLRX, GAL, 21 and CRKL, as most significantly altered proteins.
Based on the above results, they propose that the comparison between CSF and mDA neuron culture supernatant allowed us to successfully identify specific secretome changes in GBA-PD attributed to the mDA neuronal population.
Although the role of FKBP4, THOP1, GLRX, GAL, 21 and CRKL on GBA-PD patients remains unknown, I think this work is well organized and a nice first contribution to the importance of the secretome analysis on both CSF and the supernatant of iPSC-derived mDA neurons of GBA-PD patients.
I have some concerns to address as below.
Major points:
- I would not think FKBP4 is the most significantly altered protein in the CSF analysis (lane 164), What data did the authors use to make that decision?
Response: We evaluated the changes in GBA1-PD CSF proteins using multiple t-tests with the 5% false discovery rate correction for multiple comparisons. As indicated in Supplementary Table S3, FKBP4 is the most significantly altered protein in the CSF analysis with a p-value of 0.000641 and a q-value of 0.020800.
- Why are the six proteins, TXNDC5, SEMA3F, APLP1, LRP11, CD164, and NPTXR, detected in CSF secretome in GBA-PD patient not detected in the supernatant of GBA-PD iPSC-derived mDA neurons? I would think this was due to the difference in sensitivity in both analyses.
Response: We consider that some changes in proteins detected in CSF, but not found in the supernatant of iPSC-derived mDA neurons, were driven by different cell types than dopaminergic neurons, such as glia and other neuronal populations. Conversely, the identified overlapping proteins in CSF and the supernatant likely represent the alterations attributed to the mDA neuron population. This is now discussed in lines 285-288.
Reviewer 3 Report
Comments and Suggestions for Authors
Authors analyze the levels of metabolism-linked protein in the CSF and mDA neuron culture supernatant from 17 GBA-PD patients and 17 iPD. This manuscript has some value to be published in IJMS, but there are some specific comments.
Comments to the authors.
Major comments:
Authors identified that the level of FKBP4, THOP1, GLRX, GAL, and CRKL in the CSF and mDA neuron culture supernatant from 17 GBA-PD patients and 17 iPD. I think that it is necessary to consider the interaction of these proteins in GBA-PD.
Kaiser et al. identified six differentially expressed proteins in the CSF of GBA-PD patients, which indicates increased neuroinflammatory events linked to the loss of dopaminergic neurons in GBA-PD. IN this study, authors emphasize some immune system abnormality (mainly FKBP4) in GBA-PD. How do authors explain the different results between Kaiser’ paper and this manuscript?
How do authors investigate to clarify whether the proposed immunological alterations are specific to GBA-PD or represent a general phenomenon in PD?
Minor comments:  
Authors find five proteins in CSF from GBA=PD. In the title, they emphasize only “FKBP4”.
Are other four proteins not so important? Reconsider the title or the end of abstract.
Author Response
Responses to the comments from Reviewer III
Authors analyze the levels of metabolism-linked protein in the CSF and mDA neuron culture supernatant from 17 GBA-PD patients and 17 iPD. This manuscript has some value to be published in IJMS, but there are some specific comments.
Comments to the authors.
Major comments:
Authors identified that the level of FKBP4, THOP1, GLRX, GAL, and CRKL in the CSF and mDA neuron culture supernatant from 17 GBA-PD patients and 17 iPD. I think that it is necessary to consider the interaction of these proteins in GBA-PD.
Response: Since GBA1-PD is clinically- and pathophysiologically- almost indistinguishable from idiopathic PD, it is challenging to associate the roles of identified proteins with the clinical presentation or pathogenic mechanisms of GBA1-PD. Our study primarily aims to discover GBA1-specific alterations, which provide insights to elucidate the molecular mechanisms underlying GBA1-PD. Although it is beyond the scope of this paper to clarify the precise interactions of the identified proteins in GBA1-PD, combining our findings with previous reports leads us to hypothesize that GBA1 mutations might influence the immune system, which potentially involves some of the identified proteins such as FKBP4.
Kaiser et al. identified six differentially expressed proteins in the CSF of GBA-PD patients, which indicates increased neuroinflammatory events linked to the loss of dopaminergic neurons in GBA-PD. IN this study, authors emphasize some immune system abnormality (mainly FKBP4) in GBA-PD. How do authors explain the different results between Kaiser’ paper and this manuscript?
Response: The most notable difference between Kaiser’s paper and our study is that while they compared GBA1-PD with unaffected GBA1 mutation carriers, we investigated the difference between GBA1-PD and idiopathic PD. Besides, their identified proteins, CALCA, CD2, DLK1, GCH1, IL17A, and SEMG2, were not included in our analysis; hence, we could not directly compare their study and our findings. Although our primary focus was investigating metabolism-related alterations, the identification of FKBP4 as the most significantly altered protein indicated the association of GBA1 with the immune system, connecting our findings with Kaiser’s paper.
How do authors investigate to clarify whether the proposed immunological alterations are specific to GBA-PD or represent a general phenomenon in PD?
Response: To evaluate the impact of the GBA1 mutation exclusively, we employed GBA1-PD patient-derived iPSCs and their isogenic control cells in our analysis. Given that the GBA1 mutation was the sole difference between the mutant and isogenic cells, the identified CSF proteins, validated in the supernatant, strongly suggest that the observed alterations are directly attributed to the GBA1 mutation. However, based solely on our data, it was not reasonable to rule out the possibility that the identified secretome changes are shared among GBA1 mutation carriers irrespective of PD status. Therefore, our findings of the GBA1-PD specific alterations need to be further verified with additional controls, such as unaffected GBA1 mutation carriers and unaffected healthy controls. It will be beneficial to utilize healthy control iPSCs and GBA1-mutation-introduced iPSCs in addition to GBA1 mutant/isogenic iPSCs to investigate the GBA1-PD specific immunological alterations at molecular levels.
Minor comments:  
Authors find five proteins in CSF from GBA=PD. In the title, they emphasize only “FKBP4”.
Are other four proteins not so important? Reconsider the title or the end of abstract.
Response: We identified FKBP4 as the most significantly altered protein in the CSF analysis, and FKBP4 was also found in the analysis of the supernatant from iPSC-derived mDA neurons. Reduced intracellular FKBP4 level in GBA1-PD iPSC-derived mDA neurons was confirmed by Western Blot, reinforcing the significance of FKBP4. We have now changed the last sentences of the abstract.